# Using core components in process evaluation: Passport skills for life

**Melek Alemdar** [1,2]*, **Pamela Qualter**[2], **Michael Wigelsworth**[2], **Annie O'Brien**[2], **Suzanne Hamilton** [2], **Neil Humphrey**[2]

**1** Ahmet Keleşoğlu Faculty of Education, Necmettin Erbakan University, Konya, Türkiye, **2** Manchester Institute of Education, The University of Manchester, Manchester, The United Kingdom

\* melek.alemdar@manchester.ac.uk, melek.alemdar@erbakan.edu.tr

## Abstract

Although school-based SEL programmes show robust effects in controlled studies, their real-world impact varies due to inconsistent implementation. Core components (CCs), the essential theory or evidence-based elements required to reliably produce intended effects, offer a framework to address this gap. A CC-informed approach can shift evaluation from whether a programme was delivered to how its active ingredients functioned in context and how well. We conducted a two-phase qualitative process evaluation of Passport: Skills for Life, integrating top-down and bottom-up evidence. Phase 1 involved systematic analysis of manuals, lesson plans, and resources to distil practice and instructional CCs. Phase 2 examined enactment via non-participant observations (n = 12) and semi-structured teacher interviews (n = 9) across four primary schools in North-West England. Data were analysed using deductive content and thematic analyses. We identified five practice and seven instructional CCs. Core practices—coping, self-awareness, and social awareness—were typically delivered, whereas more complex interpersonal skills (help-seeking/giving, relationship skills) were seldom observed. Instructionally, low-intensity formats (didactic talk, whole-class discussion, short written tasks) predominated, while high-engagement activities (role-plays, situation cards, Dragon's Path) were often truncated or omitted. Delivery quality varied: facilitation sometimes built conceptual understanding, competence-building promoted rehearsal and modelling, yet disengaged practices weakened socio-emotional norms. Teacher warmth, responsiveness, and emotional literacy were pivotal; when present, SEL activities reinforced prosocial behaviours, but when absent, delivery narrowed to surface-level coverage. By linking programme theory with classroom enactment, this study shows how a CC-informed lens can reveal fidelity and quality of delivery patterns, offering a theory-aligned framework for scaling and sustaining school-based SEL.

**Data availability statement:** The data underlying this study are stored on secure, encrypted servers held by the University of Manchester. Due to ethical restrictions imposed by the University of Manchester Research Ethics Committee (Ref: 2022-14050-2440), the qualitative interview and observation data cannot be made publicly available, as they contain potentially identifiable and sensitive information relating to participants. Data access requests should be directed to the University of Manchester Office for Open Research (open-research@manchester.ac.uk), which serves as the institutional contact for external data access requests. Requests will be reviewed in accordance with institutional data governance and ethical requirements. Access to available anonymised data may be granted through researcher collaboration with the University of Manchester. To support transparency and reproducibility, the study codebooks are publicly available on OSF: https://osf.io/uh7yt/files/osfstorage.

**Funding:** The study was supported by a grant from the Kavli Trust Health Research Programme, funded by the Kavli Trust. NH secured the funding. MA is supported by the Scientific and Technological Research Council of Turkey (TÜBİTAK) through the 2219 International Postdoctoral Research Fellowship Program. The funders had no role in study design, data collection and analysis, decision to publish, or preparation of the manuscript. The views expressed are those of the authors and do not necessarily reflect the official policy or position of the Kavli Trust or TÜBİTAK.

**Competing interests:** The authors have declared that no competing interests exist.

## Introduction

As children spend over a thousand hours in school each year [1,2], schools have become key sites for promoting mental health and social-emotional development, particularly through universal social and emotional learning (SEL) interventions [3]. Passport: Skills for Life (hereafter Passport) is one such programme, designed to enhance children's emotional literacy and coping skills through structured, teacher-led activities [4]. Explicitly grounded in coping and affect regulation theory, Passport is theoretically distinctive among SEL programmes in that it targets coping flexibility through structured rehearsal of adaptive emotion regulation strategies to disrupt pathways to internalising difficulties and strengthen children's psychosocial resilience [5]. Although the developers reported promising implementation outcomes in a small-scale trial [6], our UK-based randomised controlled trial found no statistically significant effects on either primary or secondary outcomes [5]. Having ruled out implementation, programme theory, and measurement failure as primary explanations, the absence of programme effects prompted us to examine a more theoretically consequential issue: whether programme differentiation was maintained in practice—that is, whether core instructional and practice components functioned as intended when transported into routine classroom contexts. Complementary qualitative findings from our trial revealed discrepancies between teachers' self-reported fidelity, and the practices they described in interviews, including the changes or omissions to core components (CCs) of the programme [7]. CCs are the essential elements of a programme believed to generate its intended outcomes [8]. Although certain "acceptable adaptations" may enhance programme relevance and engagement [9], adaptation can be a "double-edged sword" [10] when it involves changes that compromise the intervention's CCs [11]. Also, these findings confirm the presence of positivity bias in self-reported fidelity [12] and point to differing interpretations of fidelity between teachers and researchers.

Taken together, these findings underscore the need for robust implementation assessments that integrate teacher interviews with structured observational data, capturing a more nuanced understanding of how programme and intervention CCs are enacted in practice [13]. Despite advances in identifying the CCs of SEL programmes, limited research has systematically examined how practice and instructional CCs interact with both fidelity and quality of delivery in real-world classroom contexts. Responding to these gaps, the present study applies a dual CCs framework to examine how key elements of the Passport: Skills for Life programme were implemented in English primary classrooms, focusing on both fidelity (adherence to intervention CCs; [14]) and quality of delivery (the appropriateness and calibre of delivery; [15]). This responds to broader concerns in implementation research, where fidelity and dosage are often prioritised, while quality of delivery remains underexamined despite its moderating role in the fidelity–outcomes relationship [16].

To capture what was delivered, to what extent, and how well, we combined two complementary frameworks. The SEL CCs framework [17–19] specified the practices and instructional formats theorised to drive student outcomes in Passport, while the intervention CCs framework—operationalised through Fidelity of Implementation

(FOI) [20]—illuminated the structural and pedagogical processes shaping their enactment in classrooms. Applied through both top-down analysis of programme materials and bottom-up classroom observations and teacher interviews, this dual lens demonstrates how a core components-informed approach can surface variations in delivery, contextualise fidelity and quality of delivery, and enable more nuanced evaluations of SEL implementation in everyday school settings.

## School based SEL interventions

Universal school-based SEL interventions involve the implementation of practices that equip children, adolescents and adults to competently handle challenges in everyday responsibilities and relationships [21] and improve personal development, ethical behaviour and effective working [22]. Meta-analytic and other evidence shows that SEL interventions facilitate a spectrum of intra- and inter-personal competencies [23], foster resilience to mental health problems [24], mitigate problems like bullying, victimization, and violence [25], counteract adverse processes and outcomes [26] and increase the likelihood of graduation, employment and overall health [22]. However, existing research also highlights the difficulty of implementing empirically supported programme practices in real school settings [27,28], resulting in a research-to-practice gap. The effectiveness of research-based programmes in schools is significantly influenced by the implementation quality [17], a critical variable that mediates between evidence-based SEL programmes and intended intervention outcomes [26,29] and, if maintained, ensures successful dissemination [10]. In that sense, monitoring and evaluating implementation is essential to determine the true potential of a programme [30].

## Dimensions of implementation and the symbiosis between fidelity and quality

Implementation refers to the ways in which a programme is put into practice and delivered to participants, and can be defined as "what a program consists of when it is delivered in a particular setting" [15]. High-quality implementation is pivotal for translating evidence-based programmes into effective services [8,29] and for their sound development, evaluation, and dissemination [31]. Following established guidance, implementation can be characterised along multiple, measurable dimensions: fidelity (adherence), dosage, quality of delivery, participant responsiveness, and program differentiation [32], monitoring control conditions, program reach, and adaptation [15]. Working definitions of implementation terms are presented in the S1 Table in the Appendix.

Of the eight dimensions, fidelity dominates in SEL implementation studies [29], with robust evidence supporting the moderating effect of fidelity on intervention outcomes [33]. This has led to an increasing emphasis on optimising the extent to which an intervention is delivered as planned, often overshadowing considerations of quality of delivery—that is, how well the intervention is implemented. High-quality implementation has been operationalised as practitioners' constructive implementation approaches, application of relevant skills, understanding of the programme's theoretical foundations, and their ability to deliver content confidently and enthusiastically [34–36]. Empirical findings posit fidelity and quality of delivery as both equally important aspects of implementation [14,37], and that it is the "symbiotic interaction" (emphasising that improvements in one positively affect the other, contributing to the overall success of a process or system) between both constructs that determines intervention outcomes.

Despite the above, studies systematically examining the interaction between fidelity and quality of delivery in real-world educational contexts remain limited [e.g.,16,38], contributing to the persistent research-to-practice gap. Conducting a process evaluation that explicitly addresses fidelity and quality of delivery in tandem can strengthen the internal validity of conclusions about programme effectiveness. Such an approach clarifies whether outcomes stem from features of the intervention itself or from implementation problems, thereby reducing the risk of Type III error (the inaccurate attribution of cause [39]). In a broader sense, it would assist in recognizing the elements that impact the successful adoption and implementation of SEL interventions when they are scaled up [40]. To achieve this, recent scholarship emphasizes anchoring implementation evaluations explicitly within an intervention's CCs—the essential, theory-driven elements hypothesized to drive its effectiveness [14,18,41].

## The case for a core components approach

CCs are the essential, theory- or evidence-based functions, principles, or intervention activities necessary to reliably trigger change in proximal outcomes [11,42,43]. By identifying these irreducible blocks, programmes can be analysed not as single units, but as combinations of specific techniques, strategies, or practices that drive outcomes [18,44]. Despite a history of different terms used to characterize CCs (e.g., critical components, essential components, active ingredients, or kernels of practice [17,45,46], there is a consensus that they are fundamental to successful programme implementation [8,47]

Recent work has distinguished between two types of core components in educational programmes: those that are structural—providing the framework and sequence of delivery—and those that are process-oriented, focusing on what happens in the classroom between teachers and students [48,49]. Building on this distinction, Century et al. [20] proposed a widely used framework, Fidelity of Implementation (FOI; see in S2 Table), for assessing implementation. In this way, CC identification clarifies what constitutes the active ingredients of an intervention [50], while FOI provides a structure for assessing how those ingredients are delivered in practice.

Without clearly defining and monitoring the delivery of an intervention's CCs, it becomes challenging to distinguish between intervention failure and implementation failure [16]. The lack of clear specification of a program's CCs makes it difficult to determine what exactly needs to be implemented for success, thereby complicating assessment, evaluation, improvement, and scaling efforts, and often leading to confusion among agencies and practitioners about what constitutes the essential "it" required to achieve desired outcomes [11]. For example, if the quality of implementation is high for certain components but poor for others, aggregate scores or metrics may be misleading and fail to reflect true implementation quality [51]. Even when a programme is implemented in full, the essential elements can be poorly delivered, which could adversely impact implementation quality [52]. In that regard, examining CCs in process evaluation can be useful for (1) addressing and maintaining fidelity to the underlying concepts and goals over time and across sites [11], and (2) examining potential implementation failures that may result from participants receiving substandard delivery, thus lacking the essential elements required for behaviour change [53]. Overall, it serves to reduce the risk of "throwing the baby (the new programme) out with the bath water (poor implementation)" [11].

Looking across the SEL literature, prior studies have contributed significantly by identifying and categorizing the CCs of SEL programmes [17,18]. These works highlighted common practice elements (e.g., 25 specific skills mapped onto CASEL's core competencies) as well as instructional strategies (e.g., teacher questioning, discussion-based activities), thus shifting the focus from "what works" to "what is taught and how." However, these contributions have remained largely descriptive, offering limited empirical or conceptual insight into how such components relate to implementation quality or how they might be systematically employed in process evaluations. Addressing this gap, the present study built on this conceptual foundation to identify the practice and instructional core components of the Passport programme. These components then informed a process evaluation, through which we examined both fidelity and quality of delivery using Century et al.'s [20] intervention CCs framework, FOI.

## The present study

Using a dual core-components (CCs) lens, we conducted a layered, two-phase process evaluation to examine what was delivered and how well in Passport. Phase 1 involved systematic document analysis of lesson plans, manuals, and instructional resources to specify practice and instructional CCs, drawing on established SEL CCs frameworks [17–19]. Practice codes aligned with CASEL competencies and indexed the social–emotional skills targeted in each session; instructional codes captured the range of embedded delivery strategies (e.g., discussion, role-play, writing tasks). This mapping provided the conceptual anchor for programme implementation indices.

Phase 2 assessed classroom enactment using FOI framework [20], see in S2 Table, focusing on two of its four dimensions most directly aligned with our research aims. The first, structural procedural core components (SPCC), captures the organisational "how-to" features that structure delivery, such as time allocated to activities, the sequencing of lessons, and

the prescribed use of curricular materials [46]. The second, interactional pedagogical core components (IPCC), reflects the quality of delivery by focusing on teacher behaviours and classroom interactions, including the instructional strategies used to actively engage students and the responsiveness demonstrated to their needs [20,54]. This distinction clarifies the theoretical boundary between fidelity and quality of delivery while providing a practical tool for process evaluation [55–57]. To operationalise this, we conducted systematic non-participant classroom observations (n = 12) to capture teaching practices [58], with particular attention to teacher warmth and responsiveness—constructs consistently linked to higher implementation quality [59–61]. To contextualise these observational data, we carried out semi-structured interviews with teachers (n = 9), probing the pedagogical reasoning behind observed practices and adaptations. This dual approach enabled a multi-perspective account of programme implementation [16,62].

To this end, the following research questions were formulated:

1. What are the core components of the intervention of Passport from the developer's perspective?

    a. What are the practice CCs of Passport?

    b. What are the instructional CCs of Passport?

2. To what extent did the teachers deliver the core components of the Passport as intended by the developer?

3. What observable teacher behaviours and practices were associated with the quality of delivery during the implementation of Passport?

## Method

### Design, context and participants

This study is a qualitative process evaluation embedded within the Passport to Success efficacy trial (See the main trial protocol [63]). We aimed to identify CCs of Passport and examine both the fidelity and quality of its implementation in relation to those components. Guided by the above research questions, we adopted a multiple case study design [64] and drew on both top-down (programme documentation) and bottom-up (observations and interviews) data sources to explore how the intervention was enacted in real-world classroom settings.

Participants were drawn from four mainstream primary schools in the intervention arm of a larger two-arm cluster randomised controlled trial (RCT) conducted in Greater Manchester. Schools were selected using instrumental multiple case sampling [65] to ensure variation in contextual characteristics such as school size, geographical location, proportion of students eligible for free school meals, and baseline levels of internalising difficulties. Within each school, all Year 5 teachers responsible for delivering Passport to Success were invited to participate in classroom observations and follow-up interviews. Teacher participation was therefore based on role eligibility within the trial and willingness to participate. Observations were conducted in lessons delivered by participating teachers, and interviews were subsequently carried out with those same teachers. In total, six teachers consented to take part (66.7% women, 33.3% men). All participants received an information sheet and provided written informed consent. Further details on school sampling and trial architecture are available in the registered qualitative report [7].

This study adheres to the Standards for Reporting Qualitative Research [66], see S3 Table. All study procedures were approved by the University of Manchester Research Ethics Committee (Ref: 2022-14050-2440; original approval 30 June 2022; amendment approved 6 February 2024). Observations and interviews were conducted between 13.02.2024–20.04.2024 in line with institutional ethical protocols. We followed ethical principles, including full informed consent, anonymity, and participants' right to withdraw. Data were anonymised and securely stored on encrypted University of Manchester servers with access restricted to the research team. All procedures within this research were performed in accordance with the British Psychological Society's Code of Human Research Ethics [67]. Data instruments, consent forms and coding materials are available via OSF [https://osf.io/uh7yt/files/osfstorage].

## The intervention: Passport skills for life

Passport is a universal school-based social and emotional learning (SEL) curriculum designed for children aged 9–11 years. The programme follows a structured developmental sequence and is delivered by classroom teachers in 55–60 minute sessions, with the expectation that all planned components are implemented within the allocated time [4]. Its primary aim is to equip children with a repertoire of adaptive coping strategies for managing challenging interpersonal and emotional situations. Through guided reflection and structured activities, students are encouraged to identify effective coping responses, articulate their emotions, seek help when needed, and support their peers [6].

The curriculum adopts a comic book format and integrates instructional and practice-based elements. Sessions typically begin with illustrated scenarios that prompt guided questioning and structured discussion, supporting emotional awareness and cognitive engagement with problem situations. Interactive components such as the Dragon's Path game use situation cards and role-play to provide repeated opportunities for behavioural rehearsal of coping strategies. In later modules, students generate their own scenarios, further consolidating strategy selection and application. Two overarching 'Golden Rules' frame all coping responses: strategies should improve the situation or provide emotional relief (or both) and must not cause harm to oneself or others [4]. Additional materials, including emotion cards and the Help Thermometer poster, are designed to reinforce emotional vocabulary, perspective taking, and help-seeking within a supportive classroom climate. Family information sheets and joint home activities extend learning beyond the classroom to promote generalisation of skills.

Consistent with SEL theory and the affect regulation framework, Passport is designed to enhance coping flexibility through the combination of explicit instruction and structured rehearsal [5]. Instructional components are intended to build declarative knowledge of coping strategies and emotional processes, while interactive rehearsal activities support procedural skill acquisition and strategy selection in emotionally salient contexts. Enhanced coping flexibility is hypothesised to facilitate more adaptive emotion regulation, thereby disrupting maladaptive emotional cycles associated with the development of internalising difficulties [68]. Beyond individual regulation processes, the programme's relational and prosocial content is expected to strengthen peer connectedness and reduce loneliness and bullying victimisation, contributing to broader improvements in psychosocial functioning and wellbeing [69–70]. Delivering Passport during preadolescence, prior to the consolidation of many mental health difficulties, is therefore conceptualised as a developmentally strategic approach to promoting resilience across the lifespan [71].

## Data collection

Data were collected in two phases; Phase 1 centred on the top-down identification of core components (RQ1), while Phase 2 focused on bottom-up exploration of implementation fidelity (RQ2) and quality (RQ3) through school-based inquiries.

**Phase 1: Document analysis.** To address RQ1, we systematically reviewed all programme documentation, including the teacher's manual, lesson plans, and instructional resources such as comic strips, activity guides, emotion cards, and posters. These materials, organised into 18 sequenced sessions across five modules (see S4 Table), provided detailed guidance on lesson structure, targeted SEL skills, and recommended pedagogical strategies. We coded these materials to extract both practice elements (e.g., empathy, emotion regulation, perspective taking) and instructional strategies (e.g., discussion prompts, role play, writing tasks).

**Phase 2: Observations and interviews.** To examine how core components were enacted in real-world settings, we conducted focused, non-participant classroom observations [72]. Six Year 5 teachers participated in the study, and each teacher was observed on two separate occasions, resulting in a total of twelve observed sessions across five schools. Each session lasted approximately 45–60 minutes, consistent with the intended delivery format of Passport, yielding approximately 11–12 hours of observational data.

To enhance reliability and reduce single-observer bias, the two observations for each teacher were conducted by different members of the research team. This design allowed variation in observer perspective while maintaining alignment with a shared observation protocol. Observers were familiar with the programme's structure and content and engaged in calibration discussions prior to and during data collection to ensure consistency in interpretation.

Observations were scheduled across the 2023–2024 spring term to capture variation in module content and instructional formats (e.g., discussion, role-play, games, group tasks). Although the number of observed sessions was necessarily constrained by school scheduling and trial logistics, observing each teacher twice enabled both within-teacher and cross-teacher comparison. Observational data were further triangulated with document analysis and semi-structured teacher interviews to enhance interpretive robustness.

An observation tool was developed to guide data collection, structured around the SPCC and IPCC dimensions of the FOI framework [20] (see OSF; https://osf.io/hz9nk). Due to time and access constraints, formal pilot classroom observations were not conducted; however, early observations were used to refine fieldnote structure and ensure alignment with predefined CCs. The tool combined checklist items with open-ended narrative fields. Fieldnotes systematically captured both fidelity and quality as they unfolded across key lesson segments (introduction, comic strip discussion, core activities, conclusion), following principles of "ethnography of the particular" [73]. Teachers were informed that observations were for research purposes only and not evaluative, to minimise performance bias [65].

To complement observational insights, we conducted nine semi-structured teacher interviews, each lasting approximately 15 minutes, to explore how they interpreted and implemented the programme in practice. The interview schedule, developed from the observation protocol and pilot-tested at the University of Manchester (see OSF; https://osf.io/uh7yt/files/y56zq), included prompts such as, "How do you integrate the programme materials into your sessions?" and "What strategies seem to resonate with your students?" Interviews were held in private settings, audio-recorded with consent, and covered both structural delivery and classroom dynamics.

## Data analysis

Analysis proceeded in two distinct phases, aligned with the two sources of data: programme documents and field-based materials. All authors had prior experience with this methodology, and NVivo 14 was employed for data management and analysis.

In the initial phase, programme documents were thoroughly reviewed. We used deductive content analysis [74] to derive practice and instructional core components from previous content analyses of SEL curricula [17–19]. The first author conducted this coding. Instructional element codes captured the variety of approaches utilized to deliver programme content, whereas practice element codes aligned closely with CASEL's [75] five core competencies and Passport's corresponding social-emotional skills. Activities involving multiple strategies or skills were assigned multiple codes. For instance, a typical lesson plan begins with instructions such as: "Give each child a copy of Comic Strip 8 – The Heat Is On and ask them to read it individually, as a class, or play the motion comic." This was coded as "comic strips – story/scenario providing context" and "watching multimedia (passive, e.g., video or audio clip)" and later themed under "curriculum-based materials". Similarly, the instruction "When they have all read the comic, ask them to summarize the story" was coded as "comic strips – story/scenario providing context" and "didactic instruction". Another example, "Ask what strategies the Dragon in the story could use to deal with a frustrating situation. Listen to a few coping strategies and write them on the board" was coded under "perspective taking (themed under social awareness)" and "coping with frustration (themed under coping strategies)". This phase resulted in five themes for practice elements and seven themes for instructional elements in Passport (See Table 1).

In the second phase, we conducted a reflexive thematic analysis [76] using NVivo 14, following the six-phase approach: familiarisation, coding, theme development, reviewing, defining, and reporting. This approach was selected because it emphasises researchers' interpretive engagement with data and theory while allowing a flexible balance of inductive and

**Table 1. Curriculum-derived identification of core components in Passport.**

| Deductive themes | Deductive codes | References-the entire programme | References-the observed sessions | Inductive codes |
|---|---|---|---|---|
| **Practice core components** | Accepting, seeking, offering help | 26 | 11 | Asking/giving help |
| | Coping strategies | 115 | 63 | Responsible decision making<br>Conflict resolution<br>Coping with bullying<br>Coping with rejection, abandonment or disappointment<br>Coping with unfair situations<br>Coping with change<br>Coping with frustration<br>Coping with stress |
| | Relationship skills | 22 | 15 | Communication<br>Relationship building<br>Teamwork |
| | Self-awareness | 107 | 52 | Accurate self-perception/emotional awareness<br>Emotions and body link<br>Identifying emotions<br>Understanding harmful behavior (bullying, etc.)<br>Expressing emotions/ emotional vocabulary |
| | Social awareness | 61 | 30 | Empathy<br>Perspective-taking<br>Respect for others<br>Appreciating diversity |
| **Instructional core components** | Family worksheets | 10 | 4 | Home activities |
| | Curriculum-based material | 79 | 22 | Comic strips<br>Posters or cards<br>Golden rules |
| | Didactic instruction | 105 | 50 | Teacher talk |
| | Discussions | 82 | 48 | Teachers' asking questions |
| | Games related to SEL skill | 39 | 24 | Situation cards<br>Role-play |
| | Kinaesthetic activity | 11 | 5 | Kinaesthetic activity |
| | Students' outputs | 29 | 19 | Drawing activity<br>Writing activity |

deductive coding. During familiarisation, audio files were transcribed, fieldnotes organised, and all data sources—including interviews, observations, and programme documents—reviewed in full. To develop the initial coding framework, AOB and SH independently coded approximately 20% of the dataset (a mix of observation fieldnotes and interview transcripts), after which MW reviewed and refined the preliminary thematic structure. Discrepancies were resolved through consensus in regular analytic meetings. In line with reflexive thematic analysis, no formal inter-coder reliability coefficient was calculated. Instead, the codebook evolved iteratively through team discussions, with code definitions refined, overlaps consolidated, and thematic boundaries clarified to enhance conceptual coherence. Following refinement, MA applied the finalised codebook to the full dataset, and themes and subthemes were developed, compared, and organised around central concepts. Consistent with reflexive thematic principles, we acknowledge that our theoretical positioning within SEL

and familiarity with the Passport programme may have shaped analytic attention and coding decisions. Reflexivity was embedded throughout via iterative codebook development, analytic annotations documented within NVivo, and ongoing team discussions in which assumptions were surfaced and emerging interpretations critically examined. The full thematic structure was reviewed by all co-authors to ensure analytic rigour and interpretive consistency.

We identified two themes each for fidelity and quality for both the practice and instructional core components (see Tables 2 and 3). To accurately reflect authentic perspectives and substantiate identified themes, direct quotes from interviewees and observer field notes were included. Given the small sample size, gender-neutral pronouns ('they/them') were employed during analysis and reporting to protect participant anonymity. Field notes utilized abbreviations: school ('S'), teacher ('T'), visit ('V'), observer ('O'), and interview ('I')."

## Results

### Core components of passport

We present the corresponding code frequencies for each of the practice and instructional CCs, derived from the entire curriculum and the observed sessions in Table 1. Practice CCs reflected the targeted social-emotional skills embedded across the curriculum: (1) coping strategies, (2) self-awareness, (3) social awareness, (4) relationship skills and (5) accepting, seeking, and offering help. Instructional CCs captured the pedagogical formats used to deliver the content: (1) didactic instruction, (2) discussions, (3) curriculum-based materials, (4) games related to SEL skills, (5) student outputs, (6) kinaesthetic activities, and (7) family worksheets. Student outputs usually included writing tasks, although drawing was offered as an alternative during specific sessions. 'My Reactions' worksheet was coded as the assessment and tracking theme; however, as this activity was designated as optional within the programme, it is not considered a core component.

### Fidelity to the practice core components

Findings related to fidelity to practice components highlight the extent to which teachers adhered to the programme's core SEL targets (see Table 2). Teachers frequently implemented practices designed to foster coping strategies, self-awareness, and social awareness. By contrast, behaviours related to accepting, seeking, and offering help were rarely observed, and relationship skills were typically addressed only through generic group work or peer-turn activities rather than through tasks explicitly designed for this purpose. Together, these gaps undermined the programme's secondary goal of strengthening interpersonal support and collaboration.

### Fidelity to the instructional core components

Teachers implemented selected instructional elements of the programme while making context-specific adaptations, augmentations, and omissions (see Table 2). Sessions most frequently featured teacher-led discussions and didactic instruction, consistent with the developers' intentions. Writing and drawing activities were commonly used to support coping strategies and emotional expression, whereas kinaesthetic activities were observed only occasionally, particularly in lessons on stress regulation and the link between bodily sensations and emotions.

Among curriculum-based materials, comic strips and situation cards were used most frequently. Comic strips appeared in nearly all observed sessions, with most teachers considering them effective for initiating discussions about emotions and coping strategies. However, situation cards—originally intended for role-playing in the Dragon's Path game—were rarely implemented as designed. Teachers adapted both pre-written and blank situation cards in varied ways. Some repurposed pre-filled cards as blank prompts, using them for brief discussions or writing tasks rather than the intended role-plays. For blank situation cards, use was again limited: in one "unfair situations" session, only two out of four cards were used, and students were told to "not write too much detail" and "keep it about home rather than school (S6, T1, V2, O3)" likely reflecting both time constraints and the presence of observers.

**Table 2. Fidelity to practice and instructional elements.**

| Deductive themes | | Deductive codes | References | Examples of inductive codes |
|---|---|---|---|---|
| **Practice core components** | **Adherence** | Accepting, seeking, offering help | 3 | Asking for help-giving help |
| | | Coping strategies | 75 | Responsible decision making<br>Conflict resolution<br>Coping with bullying<br>Coping with rejection, abandonment or disappointment<br>Coping with unfair situations<br>Coping with change<br>Coping with frustration<br>Coping with stress |
| | | Self-awareness | 46 | Accurate self-perception/emotional awareness<br>Emotions and body link<br>Identifying emotions/Emotional vocabulary<br>Understanding harmful behaviour (bullying,etc.) |
| | | Relationship skills | 22 | Relationship building<br>Communication<br>Teamwork |
| | | Social awareness | 21 | Empathy<br>Perspective-taking<br>Respect for others |
| | **Non-Adherence** | Accepting, seeking, offering help | 8 | Limited opportunity to facilitate asking-giving help behaviours |
| | | Relationship building | 7 | Limited opportunity for teamwork |
| **Instructional core components** | **Adherence Adherence** | Curriculum based materials | 52 | Comic strips-Story, scenario whereby context is given. Posters or cards. |
| | | Discussions | 47 | Teacher asking questions<br>Hands-up |
| | | Didactic Instruction | 60 | Teacher talk– providing explicit instruction– Not if leading discussion |
| | | Student Outputs | 33 | Writing activity<br>Drawing activity |
| | | Games related to SEL skill | 16 | Role play (interactive behavioural rehearsal component)<br>Situation cards (applied skill rehearsal tool) |
| | | Kinaesthetic activity | 8 | Kinaesthetic activity |
| | | Family pack | 2 | Home activities (rarely implemented)<br>Parent information (limited evidence of delivery) |
| | **Omissions, Adaptations and Augmentations** | Adaptations, augmentations to the components | 41 | Additional materials<br>Role play adapted (modification of interactive rehearsal format)<br>Situation cards used differently (shift toward discussion-based delivery) |
| | | Connecting with different programs | 27 | Not using Passport materials |
| | | Fully omitting or partially covering lesson component | 107 | No role-play<br>Family pack not sent<br>No conclusion/review |
| | | Changes in timing | 19 | Time wasn't enough to share more ideas<br>Time was cut short |

 

In our observations, only two teachers implemented the Dragon's Path game and consequently the associated role-plays, both with substantial adaptations. In the first classroom, the teacher omitted the role-play due to time constraints, with students instead mimicking emotions like frustration through facial expressions and gestures; while in the second, role-play was loosely facilitated using situation cards, but uneven participation, minimal teacher scaffolding, and visible emotional distress among some students diluted its intended impact. Despite the programme's emphasis on the interactive components, the other four teachers perceived the Dragon's Path game and role-plays as ineffective for their students. One teacher noted, "We don't use the stepping stone game because I know it doesn't work for my children in class. Not always doing the scenarios because…it might just bring up a wildfire of extra problems (S3, T2, V2, I1)." Another commented, "I can't remember doing [the role-play], if I'm honest with you. It wouldn't be the school layout that would stop us doing it, it would just be whether I thought it was essential within the lesson, I would say, or it's going to be effective (S3, T1, V2, I2). Similarly, the Help Thermometer was observed in just one session. A teacher explained its limited use as: "We started it and then the children kept going crazy, like 'I've helped that person!' And it just became too much of a focus (S3, T2, V2, I1)." Taken together, these patterns indicate that teachers classified, prioritised, or omitted activities according to their perceived pedagogical importance and fit with classroom dynamics. These decisions were frequently justified in interviews by reference to teachers' beliefs about effectiveness, classroom manageability, and the limits of coping instruction.

Use of supplementary resources was also limited. The family pack, designed to reinforce learning at home, was distributed by only one of six teachers. The only teacher who reported sending home Passport activities—working in a school with higher-than-average free school meal eligibility and serving predominantly minority families—noted that, while a minority engaged with the materials, overall response was limited and highly dependent on individual family circumstances: "We haven't had any questions based on those sheets that have gone home necessarily. Some of our parents aren't the best at contributing towards school life, so I think it depends on the child, depends on the parent and depends on the home life situation… Some of them don't do the everyday homework, never mind the extra sheets sent home with them (S7, T1, V2, I2)." This finding highlights the ongoing challenges of achieving meaningful family involvement, particularly in socioeconomically disadvantaged and culturally diverse contexts, and emphasises the absence of ongoing dialogue or systematic collaboration regarding SEL at home.

### Quality indicators for practice core components

Three sub-themes were identified relating to quality indicators for the practice elements: facilitating SEL skills, competence building: practise promotion and modelling, and undermining SEL processes. These capture differences in how effectively the programme's practice elements were enacted. An overview of these themes and their sub-codes is provided in Table 3.

**Facilitating SEL skills.** This theme refers to instances where teachers clarified and contextualised SEL concepts, encouraged students to discuss and reflect on their feelings and coping strategies, and reinforced conceptual understanding (e.g., "Teacher asking probing questions: Is this situation unfair?"). Teachers actively facilitated students' comprehension of SEL concepts by encouraging open dialogue about emotions and coping strategies, and by prompting the use of precise emotional vocabulary. As recorded in a field note:

> "The teacher asked about the emotions students experienced in the situation: "How did you feel?" and said "Thank you for putting your hands up!" T listened as students responded "envious, jealous, exhausted, sad, upset, etc." T asked, "What did you do at that point to regulate these feelings?" and offered her own example to illustrate effective emotional regulation (S15, T2, V1, O1)."

They frequently illustrated these concepts with varied examples, including personal anecdotes and real-life scenarios, to make abstract ideas more relatable. In some cases, authentic classroom situations were used to anchor discussions,

**Table 3. Quality indicators for practice and instructional core components.**

| Deductive themes | | Deductive codes | References | Inductive codes |
|---|---|---|---|---|
| **Practice core components** | **Quality indicators** | Facilitating SEL skill | 123 | Encourages students to talk about their feelings & coping strategies,<br>Encourages help seeking-accepting help,<br>Encourages the use of emotional vocabulary,<br>Presents a variety of examples, e.g., real-life experience<br>Using authentic classroom situations |
| | | Competence building: practice promotion and modelling | 37 | Active listening<br>Promoting the use of emotional literacy or coping strategies<br>Provides positive modelling of emotional expression, coping str. or helping<br>Serves as a model for students in managing emotions |
| | | Undermining SEL processes | 53 | Gender stereotyping in delivery<br>Maladaptive coping strategies<br>Normalising/allowing peer exclusion<br>Non-responsive to disclosure<br>Pre-emptive limiting of expression |
| **Instructional core components** | **Quality indicators** | | | |
| | | Facilitating student engagement | 261 | Balance between teacher and student talk<br>Effective classroom management<br>Effective feedback<br>Effective questioning<br>Encourages active participation<br>Encouraging group work<br>Encouraging peer to peer interaction<br>Prioritizing voluntary participation |
| | | Teacher warmth, support and responsiveness | 68 | Conveys interest with verbal and non-verbal cues<br>Gives personal disclosure as an example<br>Provides safe and supportive environment for meaningful discussions<br>Takes measures to prevent exclusion, disclosure of the student names |
| | | Facilitating the use of materials, manipulatives, and tools | 40 | Effective and active use of games<br>Effective use of comic stories<br>Reviewing the previous session |
| | | Differentiated instruction | 29 | Alternative strategies for specific needs |
| | | Pedagogical Disengagement | 168 | Lack of involvement in students work<br>Minimal effort to teach coping strategies<br>Not creative enough to engage active participation<br>Poor time management<br>Poor feedback<br>Shortcomings in facilitating use of materials and tools<br>Not encouraging quiet ones<br>Just QA not teaching coping strategies<br>Fast paced talking- not enough time for children to speak<br>Stifles expression<br>Lacking warmth and responsiveness |

enabling students to connect SEL principles directly to their own experiences. These practices helped bridge conceptual understanding with everyday application. As a field note reflected:

> "Teacher summarises so far what they have looked at and asks what about coping? How do we cope? Writes coping with change on whiteboard. Hands up. For examples given T asks more probing questions or provides examples of when they do these coping strategies together as a class or in assembly. Referring to the time they went to the woods teacher asks 'did it help you feel better during this time? Who came into class feeling better?'(S15, T1, V2, O2)."

**Competence building: practice promotion and modelling.** This theme describes instances where teachers explicitly demonstrated SEL skills, guided students through practical application, and practised active listening (e.g., selecting a student to role-play frustration and prompting applause). Teachers frequently modelled emotional expression and coping strategies, drawing on both real-life and classroom-based scenarios to illustrate how students might apply SEL skills in practice. Competence building was fostered through active listening, promoting the use of emotional literacy and coping strategies, and providing positive models of emotional expression and helping behaviours. For example, teachers deliberately demonstrated active listening as part of their instructional approach: "The classroom is quiet; T lowers his/her voice so as not to disturb the students, models 'how to listen in an effective way' and uses body language, offering different perspectives on the situations (S15, V1, T2, O2)."

Teachers explicitly positioned themselves as role models for managing emotions, making discussions personally relevant by sharing their own experiences. As mentioned in an interview: "To make it personal and relatable… I try to give children examples of things they've seen me do. They've been in my class when I've had to deliver a staff meeting… I'm 100% subconsciously modelling… being stressed, but that's what they need to see. If I didn't give them an example of when adults are stressed, it would be hard for them to understand… (S3, T1, V2, I1)."

**Undermining SEL processes.** This theme captures delivery characterised by minimal investment or enthusiasm, reducing opportunities for meaningful skill development (e.g., rushing through a lesson, with most time spent on conflict experiences and strategies that focused on exclusion). Observational data revealed instances where instructional practices not only lacked clarity or effectiveness but also risked undermining the establishment of positive SEL norms. For instance, insensitive teacher responses to students' emotional disclosures or dismissive remarks during sensitive discussions were noted. In one example, a teacher reacted to a student's contribution by remarking, "That's just an opinion (S6, T1, V1, O1)" potentially discouraging further participation. Another instance from the same teacher illustrated a superficially positive yet emotionally detached response "That's a good one" acknowledging the student's attempt without addressing the emotional content of their disclosure.

Some notes reflected a more overt contradiction between the programme's intended climate of inclusion and the messages conveyed to students. When a child expressed willingness to play with everyone, the teacher replied, "It's a nice idea but just not realistic. Sometimes you just get sick of other children. It's natural. Don't force it" (S6, T1, V1, O2)—a statement that normalised social exclusion and discouraged prosocial initiative. However, the same teacher positioned impartiality as their primary pedagogical principle in the interviews: "I try to avoid giving too much opinions and more just sort of trying to present the facts… some of the issues are quite sensitive and it's opinion-based… it's trying to make sure that it's not opinions" (S6, V2, I2). A closer look at the interview data reveals that the teacher does not consider the teaching coping skills as a core part of the curriculum. Instead, they believe that it is important to teach children that not every situation can be resolved by using a coping strategy: "I think sometimes they think that everything's got a solution, but I think as they get older, they'll sort of realise that some things are unfair and there isn't always an answer to it. So, just trying to sort of give them this idea of…sometimes you can't fix everything (S6, V2, I2)."

## Quality indicators for instructional core components

Here, we examine the quality of instructional core components, which are the observable features of teaching through which curricular intent becomes student experience. We organise findings from systematic classroom observations around practice domains that either enable or impede learning opportunities: (a) facilitating student engagement; (b) teacher warmth, support, and responsiveness; (c) facilitating the effective use of materials, manipulatives, and tools; and (d) differentiated instruction and (e) pedagogical disengagement. Taken together, these themes capture the presence or absence of effective teaching strategies, enabling us to characterise variations in quality within and across lessons, and consider their implications for implementation. An overview of the themes and sub-themes is given in Table 3.

**Facilitating student engagement.** Observations revealed that actively facilitating student engagement with both lesson content and peers was a defining feature of effective practice. This engagement was cultivated through effective questioning, strategic classroom management and targeted feedback. Questioning was used to encourage critical thinking and personal connection, not simply to test recall, while feedback reinforced effort and sustained involvement. For example:

> "Teacher asks children to summarise. "What does this mean? Hands up". Children in pairs attempt to summarise the story to one another. T goes around to groups to check in and asks questions about story. 'What were they doing? Who couldn't go? That's right, well done. Can you remember why? How do you think they felt? How do you feel when you're told you're too young? They were feeling… What did they decide to do? A sol…yes! A solution! What's a solution? Excellent!' (S15, T1, V1, O1)."

Collaborative activities, such as working in pairs or groups, gave students more opportunities to relate SEL themes to their own experiences and practise pro-social behaviours. Although voluntary participation was encouraged, teachers also employed adaptive strategies to ensure inclusion. As one teacher explained: "Every child is important, every child's voice matters… sometimes I'll question tables, sometimes a child, sometimes we'll do a wave because I'm conscious that sometimes children aren't contributing… and take the lesson where the children take it" (S15, T2, V2, I1).

**Facilitating the use of materials, manipulatives and tools.** Observational data highlighted the importance of effectively using materials, manipulatives, and instructional tools to enhance student engagement and learning. High-quality implementation of these resources was consistently observed across different contexts. Specifically, comic strips emerged as particularly effective in introducing key emotions and coping strategies, as well as modelling essential social competencies. Teachers frequently paired comics with reviews of coping strategies discussed in previous sessions, allowing students opportunities to reflect and deepen their understanding:

> "Teacher pauses during comic animation to ask children what could be happening, "Lots of adventurous guesses". Teacher pauses and asks children 'Can you tell how she's feeling?' Asks how the characters feel and why, and how do we know this. Ties it to previous lesson when T hear about coping strategies they covered (S15, V2, T1, O1)"

In some sessions, teachers effectively incorporated authentic classroom situations to make the content more relatable and meaningful. Examples included referencing recent events such as a basketball game or a morning conflict, modelling yoga moves from past activities, and revisiting experiences such as an outdoor walk to link them to coping strategies:

> "Teacher asking probing questions - thumbs up or down - is this situation unfair? T ties it into workshop yesterday on visit from disability activist. Gives examples of unfair versus unequal situations on children feedback on it via hands up. Reflected on coping strategies used in comic. Children finished teacher's sentences (S15, V1, T2, O2)"

**Teacher warmth, support and responsiveness.** Observational data frequently highlighted teacher behaviours related to warmth and responsiveness. Teachers commonly demonstrated warmth by using encouraging language, facilitating

open dialogue, and actively fostering supportive interactions among students. Interest was conveyed through verbal and non-verbal cues such as smiling, making eye contact, using encouraging language and acknowledging contributions with specific praise. This attentiveness fostered a classroom climate in which students felt comfortable sharing and engaging. Although situation cards were being used in an adapted way in a role play, an observer noted: "Teacher is warm, children are engaged and there is a feeling of support. Teacher is sensitive and empathetic, and this creates a nice atmosphere in which children appear to feel comfortable sharing and are engaged and listening (S7, V1, T1, O3)."

Teachers used personal disclosures strategically to establish relatability and strengthen connections with students. By sharing relevant personal experiences, they created a more approachable and engaging atmosphere. A teacher explained, "Especially in Passport, a lot of the times if I've said about emotions, I'll link an example of where that emotion was linked to something I've done. And I think they like that because it makes them feel like I'm more human with them and I make mistakes like them and things as well (S7, V1, T1, I1)." One example we observed (in session for dealing with emotions that comes from loss) was a teacher who shared a story about adjusting to losing a pet: "I had a hard time adjusting when we got a new dog (S15, V2, T2, O2)."

Teachers also implemented deliberate strategies to create safe spaces for sensitive discussions. This included anonymity protocols, such as encouraging students to adopt pseudonyms (e.g., "Dave" or "Sue") when recounting experiences of unfair treatment, thereby protecting privacy and reducing fear of judgement. Similarly, one teacher facilitated a conversation about stress management by asking students to indicate anonymously how well they coped: "Teacher says 'Heads down and hands up if you deal well/ if you don't deal with stress well.' adds "there can be differences in what others think versus how you actually feel you manage stress (S3, V1, T2,O1)."

**Differentiated instruction.** Another important quality indicator identified through observational data was differentiated instruction, which played a key role in meeting the diverse needs of pupils during SEL sessions. For example, during five sessions, teaching assistants provided targeted support for pupils with special needs (e.g., for a pupil with obsessive-compulsive disorder). In one session, a teacher provided one-to-one support to a pupil from a different ethnic background, helping him to articulate his thoughts about major life changes, such as changing schools and learning a new language.

However, a teacher highlighted the challenges associated with the program's lack of built-in differentiation. To address the gaps, the teacher described employing pre-teaching strategies to equip pupils with the vocabulary needed to express themselves effectively. As the teacher explained, "Because they're not very differentiated, I think as teachers ourselves, we differentiate it for our class rather than... it's there. (S6, V2, I2)".

**Pedagogical disengagement.** Observational data revealed instances of pedagogical disengagement, characterized by limited encouragement of active student participation, minimal teacher involvement in student tasks and emotionally detached communication. This theme was also indicating that teachers occasionally relied on routine instructional methods, failing to capture student interest or stimulate meaningful interaction.

Lack of involvement in students' work was a recurrent theme. As mentioned previously, only two teachers were observed using situation cards for role play. In one case, both the preparation and enactment phases were dominated by logistics rather than pedagogical engagement. During preparation, "Different children are then moved to different groups. Teacher says to move into your corner so that the teacher can figure out who has been moved and who has yet to be moved. Aggressive and condescending tone used by teacher towards children and the quiet children move back into their far corner (S6, V1, T1, O1)." In the enactment—captured by a different observer—the teacher's attention remained on paperwork rather than students' work: "Teacher is looking at the curriculum, not the role play! T moved a student to another place… The group is acting but the T isn't watching" (S6, V1, T1, O2).

The minimal effort made to teach coping strategies, coupled with an over-reliance on Q&A without explicit instruction, further limited learning opportunities. In one lesson, which began with a brief recap after a four-week break, "Recap to last lesson (4 weeks ago). 'I know we haven't done passport for a while but our visitors would like to see it…(S3, V2, T1, O3)" In another case, a behaviour incident was met with reprimand rather than guidance: "It's embarrassing! You should be

ashamed of yourselves (S6, V1, T1, O2)." Interestingly, when interviewed, the same teachers identified conflict resolution as their primary strategy for dealing with student disputes. As the teacher explained: "Conflict resolution, pulling them all to one side…[I] give them a rationale perspective of the bigger picture, of why it's not really that important or…their worry isn't that sort of prominent (S6, V2, T1, I2)."

Some lessons were not creative enough to engage active participation or failed to encourage quieter students, relying on predictable formats. Fast-paced delivery constrained student voice: insufficient pauses left children with little time to think or contribute, stifling expression and narrowing discussion depth. Poor time management often compounded these issues. One session consisted primarily of whole class listening to a small set of scenarios, with deeper work deferred: "Lesson ended after 4 (?) situations had been read out. Pupils seemed to enjoy this, but it was a whole class activity of listening. T said they would read more out the next day if they have time (S3, V2, T1, O2)." In another observed lesson, extended focus on definitions left insufficient time for deeper discussion: "Everything was very surface level… By the time the lesson progressed to more meaningful issues of why people bully, time had run out… The teacher appeared quite anxious about having the children finished by a certain time so they could do PE (S3, V2, T2, O3)."

Finally, lacking warmth and responsiveness was a consistent pattern in disengaged lessons. A teacher was observed raising his voice to issue abrupt instructions, like "2 minutes". This tone was mirrored in responses to pupils' affective disclosures, as one observer noted: "The teacher did not validate their feelings or experiences (actually did the opposite—dismissed) (S6, V2, T1, O3)" The second observer described this stance as emotional unavailability toward pupils' disclosures (S6, V2, T1, O2).

## Discussion

Although the promise of SEL is well established, its impact depends on how programmes' core components (CCs) are enacted in classrooms [18,41,55]. To open the "black box" of implementation, we used a dual CC framework and triangulated programme materials with lesson observations and interviews. This approach revealed uneven implementation: while lower-intensity targets and formats were consistently delivered, complex interpersonal skills and engagement-rich methods were often truncated.

Importantly, practice and instructional CCs proved interdependent: facilitation, competence building, and erosion of SEL processes unfolded through instructional mechanisms such as engagement, responsiveness, use of materials, differentiation, and, at times, disengagement. These findings suggest that the "what" of SEL cannot be separated from the "how" of its pedagogy, and that impact hinges on the dual commitment to explicit instruction and the socio-emotional climate in which skills are rehearsed and reinforced [16].

Taken together, our findings point to a layered conceptual pathway: SEL impact depends on how core components are enacted; enactment varies in intensity and depth; practice and instructional components operate interdependently; instructional mechanisms (e.g., engagement, responsiveness, use of materials, differentiation) mediate their activation; and these processes collectively shape the socio-emotional climate in which skills are rehearsed. Within this pathway, educators' own SEL competencies emerge as a critical determinant, influencing whether core components are enacted superficially or in ways that enable meaningful competence building and transfer [77,78].

### Clarifying core components of passport

Through RQ1, we were able to systematically identify the core components of Passport. Document analysis identified five practice CCs—coping strategies, self-awareness, social awareness, relationship skills, and accepting, seeking, and offering help—and seven instructional CCs—didactic instruction, discussion, curriculum-based materials, SEL-related games, student outputs, kinaesthetic activities, and family worksheets. This dual categorisation captures both the social-emotional skills the programme seeks to cultivate and the pedagogical strategies through which they are delivered.

Although the CCs literature within SEL is relatively sparse, the limited studies to date have advanced two main approaches. Descriptive frameworks have mapped broad categories of practice and instructional strategies [17,18,44], clarifying what CCs might look like in theory but offering little empirical leverage on their role in implementation quality or process evaluation. Empirical studies, meanwhile, have positioned CCs primarily as predictors of student outcomes. Molloy et al. [79], for example, identified active ingredients of Positive Behaviour Interventions and Supports statistically, modelling which elements predicted variation in behavioural outcomes. Abry et al. [50] selected a subset of Responsive Classroom components based on programme theory and prior evidence, linking teacher-reported and observed fidelity indices to student achievement. By systematically identifying Passport's practices and instructional CCs and then examining their implementation in real classrooms, our study offers a third perspective: treating CCs as tools for evaluating implementation quality.

## Unpacking the Black Box: Core components and contextual enactment

Findings relating to RQ2 reveal a mixed fidelity profile. Several practice CCs—particularly coping strategies, self-awareness, and social awareness—were delivered in line with programme intentions, suggesting that Passport's primary SEL targets were at least partially preserved. However, secondary objectives involving more complex interpersonal behaviours such as help-seeking, help-giving, and relationship skills, were rarely observed. The limited enactment of help-seeking and help-giving behaviours is particularly salient, as it reflects a design challenge that the programme developers already recognised. In earlier iterations of Passport, this gap prompted the inclusion of the Helping Thermometer within the Dragon's Path game as a targeted mechanism to model and practise these behaviours, and its importance was reiterated during mid-programme training [6]. Yet, in our trial, Dragon's Path was rarely implemented as intended, and the Helping Thermometer was seldom used. By omitting or simplifying these structured, game-based activities, teachers reduced opportunities for sustained, collaborative practice of relationship skills—replacing them with low-intensity interactions (e.g., turn to your peer) that likely fall short of the programme's intended learning mechanisms.

A similar gradient emerged for instructional CCs. Lower-intensity formats—didactic instruction, whole-class discussion, and short written tasks—were generally delivered as intended, whereas high-engagement formats—role-plays, situation cards, Dragon's Path, and family worksheets—were frequently truncated, integrated into other SEL initiatives, or omitted. Three recurrent adaptation patterns were identified: (1) omission or partial delivery of signature activities, (2) integration of Passport content with other SEL programmes, and (3) alterations to timing, including prolonged pauses in delivery. In practice, teachers classified, prioritised, or omitted activities based on perceived importance and pupil need/engagement.

One of the most pronounced omissions concerned family engagement. Only one teacher reported sending home activities, and there was no evidence of sustained home–school collaboration. This pattern aligns with sector-wide data: in a review of 252 universal, school-based SEL interventions, 40.8% reported no family component; among those that did, only 5% provided at-home materials [26,34]. The omission matters: meta-analytic and empirical evidence links family–school approaches to gains in children's prosocial skills, peer relationships, and mental health [80–83]. Consistent with ecological theory [84] and CASEL's systemic SEL framing [85], authentic home–school partnerships are the mesosystemic bridges that reinforce classroom learning.

Interview data suggest why these bridges are hard to build. The only teacher who enacted the family component—working in a socioeconomically disadvantaged, culturally diverse context—described participation as inconsistent and highly contingent on family circumstances. While teachers sometimes perceive limited reinforcement at home as a barrier [86,87], research indicates that low engagement often reflects unclear communication rather than parental disinterest [88]; across backgrounds, families value education and want to be involved [89]. Notably, meta-analytic findings show that regular, meaningful home–school communication (notes, email, text) yields the largest gains in social–emotional competence (ES = 0.66) and mental health (ES = 0.64) among family–school components [83]. Absent these structures, opportunities to consolidate SEL learning beyond the classroom are lost.

 

In our broader programme evaluation, intention-to-treat analyses showed no detectable effects of Passport on any measured outcomes. Having dismissed the implementation failure, programme theory failure, or research failure, we identified differentiation failure—i.e., erosion of treatment contrast—as the most plausible explanation [5]. The present findings help specify how this erosion occurred in practice. As evidenced by the documented pattern of omissions and adaptations summarised in Table 2, key interactive components—such as role-play activities and family-extension tasks—were frequently omitted or substantially modified, with delivery often converging on lower-intensity formats (e.g., whole-class discussion, brief written tasks). This observable shift away from structured interactive elements likely reduced opportunities for behavioural rehearsal and home–school linkage embedded within the programme design, thereby limiting its potential to produce measurable change. These patterns are consistent with wider evidence that effects diminish when moving from efficacy trials to routine delivery [90] and with arguments that variability in implementation is a key driver of such attenuation [14].

## Not Just What, But How: Unpacking the Symbiosis

Across classrooms, delivery quality was most evident when facilitation of SEL skills and competence building intersected with three instructional mechanisms: student engagement, effective use of materials, and differentiated instruction.

Facilitation of SEL skills was the most prevalent practice, as teachers prompted students to name emotions, identify coping strategies, and connect content to classroom life. These practices primarily fostered declarative knowledge—facts, labels, and steps [91]—through narration and explicit modelling [92]. Yet declarative knowledge alone is insufficient for transfer [93]. Durable learning requires scaffolding that moves students toward procedural and conditional rehearsal through repeated enactment and guided practice across contexts [90,94], thereby creating the sustained opportunities necessary for generalisation [95,96].

Competence-building opportunities, though less frequent, proved pedagogically consequential. When teachers modelled coping strategies, facilitated active listening, or promoted emotional expression, they provided moments of procedural rehearsal—and occasionally conditional use—that deepened skill acquisition. Crucially, these opportunities consolidated only when coupled with sustained engagement and active rehearsal. Balanced teacher–student talk, purposeful questioning, positive feedback, and structured peer interaction moved students beyond recall toward rehearsal, consistent with evidence that student-centred pedagogy embeds competencies rather than merely "covering" content [97,98].

The use of materials and tools further shaped these processes. While didactic instruction, songs, and handouts may reinforce declarative knowledge, role-plays and games are essential for skill generalisation and transfer [17]. Declarative knowledge was strengthened when students conceptually understood what a material—such as a comic strip or situation card—represented, whereas procedural knowledge emerged only through active enactment using these tools. Although Passport was explicitly designed to embed high-engagement formats [6], teachers often replaced them with whole-class discussions or omitted them altogether, stripping away mechanisms—repetition, modelling, rehearsal—that consolidate procedural knowledge. From the Universal Design for Learning [99] perspective, this shift represents a narrowing of the programme's multimodal and engagement-rich design, reducing opportunities for joyful, embodied rehearsal and multiple means of participation that support inclusive learning environments.

Finally, differentiated instruction—though less frequently observed, as teachers noted it was not sufficiently built into the programme [7]—served to amplify quieter voices, adjust task demands, and ensure equitable access to competence-building opportunities. Such tailoring aligns with equity-centred SEL [73] and underscores that competence building consolidates only when supported by relational sensitivity and adaptive pedagogy.

Taken together, method choice may function as a proxy for teachers' pedagogical beliefs, instructional routines and habitus, and classroom management capacity, thereby mediating whether components are enacted deeply enough to yield transferable competencies. As others have noted programme success hinges less on design than more on quality of delivery [35,100].

## Erosion of SEL processes, responsiveness, and pedagogical disengagement

Across classrooms, delivery quality faltered most visibly when relational supports were absent, and pedagogy slipped into disengagement—patterns that eroded the socio-emotional climate essential for competence building. The third instructional CC pattern—erosion of positive SEL processes—surfaced when delivery practices undermined the classroom climate required for skill development. Episodes included poor feedback, promotion of maladaptive coping (e.g., avoidance), normalisation of peer exclusion, and teacher non-responsiveness to student disclosures. Such practices constrained emotional expression and weakened socio-emotional norms. Crucially, the same instructional element (e.g., situation cards) could either constrain or cultivate the socio-emotional climate depending on delivery. In one classroom, enactment was dominated by logistics and teacher detachment, hollowing out socio-emotional potential; in another, teacher warmth and sensitivity transformed the identical task into a collaborative experience that fostered emotional support and, in turn, prosocial engagement [101].

Instructional CCs emphasise two dynamics that magnified this erosion. First, teacher warmth, support, and responsiveness acted as a bridge between reinforcement and erosion. Warm and attentive teachers fostered an environment conducive to open emotional expression [102], whereas their absence rendered activities superficial or even detrimental. Second, pedagogical disengagement occurred when curricular supports such as games, comic stories, and narrative reviews were either poorly used or reduced to mechanical delivery, thereby stripping them of their interactive nature. Interview data illuminate that these divergent practices were underpinned by teachers' emotional literacy and their pedagogical beliefs about SEL—for instance, whether they framed disclosure and coping as opportunities for shared humanity or, conversely, as reminders that not every problem has a solution. This finding underscores that climate and instruction cannot be disentangled [60,101].

Ultimately, erosion is not just a technical failure, but also a failure of relational stance. Without warmth and responsiveness, modelling becomes a mechanical demonstration, and without engagement, scaffolding turns into empty rituals. Where pedagogical skill, emotional attunement and interactive responsiveness were lacking, the focus shifted from competence-building pedagogy to declarative coverage. Conversely, when teachers displayed emotional literacy, recognised student contributions and fostered a sense of connection, SEL activities promoted prosocial behaviour and enabled deeper learning. In this sense, erosion and reinforcement are two sides of the same instructional coin: facilitation and competence-building only gain traction through engagement and interactive tools, and both falter when responsiveness and pedagogical investment are absent.

Taken together, these findings highlight that quality of delivery rests not simply on whether teachers "do the programme," but on how their competencies, beliefs, and affective stance activate—or blunt—the mechanisms through which SEL aims are realised. Within this frame, educator SEL emerges as a critical determinant. Beyond technical fidelity, educator SEL encompasses the dispositions and relational skills needed to create supportive environments, integrate SEL into academic content, and model competencies through everyday interactions [76,77,94]. Effective educators make SEL visible by narrating their own use of skills—including the unlearning of counterproductive mindsets [103]—and by intentionally modelling emotional regulation and prosocial behaviours in daily exchanges [104,105]. Such practices have been consistently associated with measurable gains in students' SEL and academic outcomes [58,106,107]. Ultimately, implementation quality and programme impact hinge not only on what is taught but on how educators embody and transmit SEL in the fabric of classroom life.

## Strengths and limitations

Our triangulated approach, which combined document analysis, classroom observation, and teacher interviews, provided a balanced account of the implementation process and helped counter the known limitations of self-reporting [13]. Direct observation is often regarded as more objective and reliable than retrospective accounts by implementers, which are susceptible to recall and social desirability bias [10,12,108]. The inclusion of complementary methods contributed to reducing

mono-method bias and offered a fuller perspective on how intended, enacted, and interpreted curricula aligned in practice [62]. Taken together, this triangulated design strengthened credibility through convergence, complementarity, and critical comparison across data sources.

However, several limitations warrant consideration. First, participating teachers were drawn from schools involved in a broader trial, and participation in the qualitative component may reflect a degree of self-selection. Teachers who consented to observation and interview may have been more confident, reflective, or positively disposed toward the programme, potentially introducing selection bias.

Second, observations were time-bound and sampled from selected sessions, introducing possible reactivity and sampling effects. Although two observers, shared rubrics, and reconciliation procedures were used, the presence of researchers may have influenced teacher behaviour (observer effect), potentially encouraging more structured or guideline-adherent delivery during observed lessons.

Third, interviews were intentionally brief (approximately 15 minutes) to minimise burden in busy school contexts. While this approach facilitated participation and focused responses, it limited the depth of exploration of teachers' beliefs and contextual constraints.

In addition, teachers varied in their familiarity, experience, and confidence with the Passport programme. Differences in prior exposure and perceived ownership may have shaped patterns of adaptation or omission observed in practice.

Furthermore, student-level behavioural outcomes were not collected within this qualitative process evaluation; therefore, direct causal links between fidelity/quality indicators and student change cannot be inferred.

Finally, although Passport was originally developed in Canada and implemented in Greater Manchester primary schools, transferability to other cultural, policy, or educational contexts should be considered cautiously. Implementation processes are shaped by local norms, institutional expectations, and system-level supports, and findings may not generalise to contexts with substantially different structural or cultural conditions.

## Implications

The findings of the present study prompt some reconsideration of how SEL implementation is theorised. Specifically, our analysis suggests more nuance is needed, particularly in relation to instructional CCs. For example, we must more explicitly differentiate between two types of knowledge that are critical for learning and skill transfer: declarative (e.g., knowing what empathy is) and procedural (e.g., how to apply this concept in practice) knowledge. Our study demonstrates that it is possible to succeed at the former through basic didactic instruction and discussion, while failing at the latter due to the omission of engaging activities like role-plays and games. This implies that implementation should be understood as a fundamental cognitive and pedagogical challenge rather than simply a logistical or resourcing issue.

In terms of practical implications, preventing pedagogical disengagement and the omission of signature activities requires SEL professional development to evolve such that it addresses not only what to teach, but how to teach it with fidelity and quality. This might include provision of opportunities for teachers to develop their own social and emotional competence; development of skills in the use of engaging activities (e.g., role-plays) and strategies to manage challenges that arise when they are implemented; and, training for 'adaptive fidelity' such that CCs remain intact but other programme components can be adapted to local context/need and/or teacher preference. Elsewhere, the home/family engagement gap noted in the present study suggests a need to develop a relational engagement model that more actively seeks and values the input and participation of families in relation to SEL provision. To make this possible, resources are required, including funding that can support equitable family participation in relevant school events (e.g., transportation, childcare).

Finally, our findings raise new questions that could further advance understanding of SEL implementation. First, to build on our finding that teacher beliefs of effectiveness underpinned the omission of CCs, future research could delve deeper into the reasons why this was the case, by examining (for example) how specific pedagogical beliefs

mediate the relationship between initial professional development/training for a given SEL programme and subsequent implementation fidelity. Second, our analysis of implementation quality could be extended to a more granular level to better understand the moment-to-moment dynamics of SEL instruction, by examining (for example) the impact of pedagogical disengagement (e.g., teacher checking paperwork during an activity) on children's responsiveness. Third, while systemic issues are implied in our findings, future research could more explicitly examine their influence, including assessment of the extent to which a school's organisational capacity (e.g., leadership commitment to and prioritisation of SEL; resource/budget allocation) moderates the impact of SEL professional development/training on teachers' implementation quality and fidelity. Although student perspectives were explored in complementary qualitative work arising from the same broader trial [7], future studies could more explicitly integrate teacher and student perspectives within a single analytic framework, enabling a more holistic understanding of how implementation quality is experienced and interpreted across classroom actors.

## Supporting information

**S1 Table. Implementation dimensions.**
(DOCX)

**S2 Table. FOI of instructional materials.**
(DOCX)

**S3 Table. Standards for reporting qualitative research (SRQR).**
(DOCX)

**S4 Table. The content of the passport modules and sessions.**
(DOCX)

## Acknowledgments

The authors declare that ChatGPT-4.0 and DeepL were used solely for language editing purposes, including grammar, spelling, fluency, and reference formatting during manuscript preparation. These tools were used to enhance readability only. All conceptual development, data analysis, interpretation, and final decisions regarding the content of the manuscript were conducted independently by the authors.

## Author contributions

**Conceptualization:** Melek Alemdar, Neil Humphrey.

**Data curation:** Melek Alemdar.

**Formal analysis:** Melek Alemdar.

**Funding acquisition:** Melek Alemdar, Neil Humphrey.

**Investigation:** Melek Alemdar, Annie O'Brien, Suzanne Hamilton.

**Methodology:** Melek Alemdar, Pamela Qualter, Michael Wigelsworth, Neil Humphrey.

**Resources:** Melek Alemdar, Pamela Qualter.

**Supervision:** Neil Humphrey.

**Writing – original draft:** Melek Alemdar.

**Writing – review & editing:** Pamela Qualter, Michael Wigelsworth, Neil Humphrey.

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
