## [Decision Letter · Decision Letter 0]

10 Feb 2026

Dear Dr. Alemdar,

Thank you for submitting your manuscript to PLOS ONE. After careful consideration, we feel that it has merit but does not fully meet PLOS ONE’s publication criteria as it currently stands. Therefore, we invite you to submit a revised version of the manuscript that addresses the points raised during the review process.

We look forward to receiving your revised manuscript.

Kind regards,

Amjad Islam Amjad

Academic Editor

PLOS One

Journal Requirements:

“Funding statement

This research was supported by a grant from the Kavli Trust Health Research Programme, funded by the Kavli Trust.  NH secured the funding. [https://kavlifondet.no/en/category/projects-en/research-projects-en/programme-on-health-research-english/]

MA is externally funded by The Scientific and Technological Research Council of Turkey (TÜBİTAK) 2219 Program. [https://tubitak.gov.tr/en]

The views expressed in this paper are those of the authors and do not necessarily reflect the official policy or position of the Kavli Trust or TÜBİTAK.”

4. Please note that funding information should not appear in any section or other areas of your manuscript. We will only publish funding information present in the Funding Statement section of the online submission form. Please remove any funding-related text from the manuscript.

5. We note that you have indicated that there are restrictions to data sharing for this study. PLOS only allows data to be available upon request if there are legal or ethical restrictions on sharing data publicly. For more information on unacceptable data access restrictions, please see http://journals.plos.org/plosone/s/data-availability#loc-unacceptable-data-access-restrictions.

7. In the online submission form, you indicated that your data is available only on request from a third party. Please note that your Data Availability Statement is currently missing [the name of the third party contact or institution / contact details for the third party, such as an email address or a link to where data requests can be made]. Please update your statement with the missing information.

8. Please ensure that you refer to Figure 1 in your text as, if accepted, production will need this reference to link the reader to the figure.

Reviewer's Responses to Questions

**Comments to the Author**

1. Is the manuscript technically sound, and do the data support the conclusions?

Reviewer #1: Yes

Reviewer #2: Yes

2. Has the statistical analysis been performed appropriately and rigorously?

Reviewer #1: N/A

Reviewer #2: Yes

3. Have the authors made all data underlying the findings in their manuscript fully available?

Reviewer #1: No

Reviewer #2: Yes

4. Is the manuscript presented in an intelligible fashion and written in standard English?

Reviewer #1: Yes

Reviewer #2: Yes

Reviewer #1: This manuscript presents a qualitative process evaluation of the Passport: Skills for Life SEL programme using a core components framework. The study is clearly designed, methodologically coherent, and addresses an important implementation question in school-based SEL delivery. The two-phase approach combining document analysis with classroom observations and teacher interviews is appropriate and well aligned with the stated research questions. Overall, the manuscript meets PLOS ONE’s criteria for technical soundness. I recommend minor revisions to improve transparency, clarity, and data availability compliance

Sampling clarity

The manuscript reports observations across schools (numbers vary in sections four vs five schools mentioned). Please standardize and clarify the exact number of schools and selection criteria.

Clarify how observed lessons and interviewed teachers were selected (purposive, convenience, availability-based?).

Observer training and reliability

The paper notes that observers were trained and used a structured tool, but more detail is needed:

Duration and content of training

Whether pilot observations were conducted

Whether inter-coder or inter-observer agreement was assessed

How disagreements were resolved

Coding and analytic rigor

Please clarify:

Number of coders involved in Phase 1 and Phase 2 coding

Whether coding was double-coded

How the codebook evolved

Whether audit trails or memoing were used

A short analytic workflow diagram or stepwise description would improve transparency.

Link between data and claims

Most conclusions are appropriately cautious; however, a few interpretive statements (e.g., mechanisms of erosion of treatment contrast) would benefit from more explicit linkage to specific observed patterns or counts already available in your tables.

Consider adding a short summary table mapping each identified core component to observed fidelity patterns.

A brief limitations subsection specifically addressing observer effect and programme familiarity would strengthen balance.

If possible, add one short paragraph clarifying transferability beyond the regional context.

Reviewer #2: INTRODUCTION

Discussion about implementation challenges is a bit repetitive. I think the program differentiation concept should be deeply unfolded. Please explain why a passport is theoretically interesting, and also add one statement on the explicit knowledge gap for a powerful introduction.

Literature review

There is dense layering of theoretical background, it can make the readers confuse specially with CC and FOI.

Frameworks should be supported with diagrams to make them more understandable.

Also, compare frameworks critically.

Research questions

RQ#3 is slightly general (What teacher behaviors were important?)

Please narrow it to observable indicators of implementation quality

Methodology

Samplinh's rationale should be elaborated further.

Please discuss transferability limits as well

Intervention Description

Overly descriptive in parts

Please include the visual program theory, and also clarify the expected pathways

Data collection

There is a limited explanation of document selection

Observations and interview

The duration of observation is not that clear

Please expand the justification of the short length

Also, clarify the reliability of observer training

Data analysis

Include a reflexivity statement

Please clarify the process of intercoder agreement

Results

There is a limited interpretation of CC's dominance

A strong link between teacher beliefs should be there

There should be clear evidence of selective implementation

Add comparative analysis between schools

Also, strengthen the link between student outcomes and quality indicators

Table of evidence

Tables are dense

Few categories overlap

Please simplify tables, and it is better to add interpretive summaries below each table

Link findings to universal design for learning also

Discussion

Please simplify theoartical language, and the conceptual model should be clear

Limitations

Some limitations should be elaborated, such as teacher selection bias, short duration of interviews, and cultural generalizability.

Implications

A few recommendations are broad; please specify them and discuss system-level training requirements

For future research student perspective is missing, which should be part of it.

Ethical transparency

AI acknowledgment is comparatively brief because requirements of some journals is detailed description

.

Reviewer #1: **Yes:** sarfraz aslamsarfraz aslamsarfraz aslamsarfraz aslam

Reviewer #2: **Yes:** DR.SHUMAILA MANSHADR.SHUMAILA MANSHADR.SHUMAILA MANSHADR.SHUMAILA MANSHA

---

## [Author Response · Author response to Decision Letter 1]

4 Mar 2026

We would lıke to thank reviewers for their time and efforts.

---

## [Editor Report · Decision Letter 1]

20 Mar 2026

Using core components in process evaluation: Passport Skills for Life

PONE-D-25-50001R1

Dear Dr. Alemdar,

We’re pleased to inform you that your manuscript has been judged scientifically suitable for publication and will be formally accepted for publication once it meets all outstanding technical requirements.

Kind regards,

Amjad Islam Amjad

Academic Editor

PLOS One
---

## [Editor Report · Acceptance letter]

PONE-D-25-50001R1

PLOS One

Dear Dr. Alemdar,

I'm pleased to inform you that your manuscript has been deemed suitable for publication in PLOS One. Congratulations! Your manuscript is now being handed over to our production team.

Kind regards,

on behalf of

Dr. Amjad Islam Amjad

Academic Editor

PLOS One